# Communicating with Patients about COVID-19 Vaccination: A Qualitative Study on Vaccinators in Tuscany Region, Italy

**DOI:** 10.3390/vaccines11020223

**Published:** 2023-01-19

**Authors:** Giorgia Alderotti, Martina Felicia Corvo, Primo Buscemi, Lorenzo Stacchini, Duccio Giorgetti, Chiara Lorini, Guglielmo Bonaccorsi, Maria José Caldés Pinilla, Vieri Lastrucci

**Affiliations:** 1Epidemiology Unit, Meyer Children’s Hospital, Viale Gaetano Pieraccini 24, 50139 Florence, Italy; 2Global Health Center, Meyer Children’s University Hospital, Viale Gaetano Pieraccini 24, 50139 Florence, Italy; 3Medical Specialization School of Hygiene and Preventive Medicine, University of Florence, Viale GB Morgagni 48, 50134 Florence, Italy; 4Department of Health Sciences, University of Florence, 50134 Florence, Italy

**Keywords:** COVID-19, vaccine, vaccination campaign, health communication, communication principles, communication strategies, vaccine hesitancy, vaccine confidence, vaccinators

## Abstract

The rapid development of the vaccine and the infodemia have challenged communication about COVID-19 vaccines. This study aims to characterize—through the experience of vaccinators—the challenges faced during COVID-19 vaccination consultations and the communication strategies adopted. A qualitative study was conducted on COVID-19 vaccinators in Tuscany, Italy. Face-to-face interviews were conducted and examined by thematic analysis. In total, 30 vaccinators were interviewed. Four main themes emerged. The first highlighted distinct profiles of users’ attitudes toward COVID-19 vaccination. Barriers and promoters of vaccine uptake emerged in the second theme: concerns over the vaccine, excessive exposure to information, and a lack of clear guidance from institutions were the main factors behind hesitancy. The third theme highlighted users’ information-seeking behaviors; vaccinators observed that users ideologically opposed to the vaccine (IOV) unconsciously seek information that confirms their theories. The last theme comprised communication strategies for dealing with hesitancy. Empathy, first-hand examples, transparency, and tailored communication style appear to be effective in building vaccine trust. Lastly, the impossibility of developing a decision-making partnership with IOVs was noticed. These findings may help to better characterize public attitudes toward COVID-19 vaccination and highlight key communication principles and strategies to foster vaccine confidence.

## 1. Introduction

Vaccination is the main tool for the primary prevention of infectious diseases and among the most cost-effective public health measures available [1]; the immunization of the population not only reduces the spreading of diseases and prevents people from severe consequences of the diseases but also considerably reduces public spending on health [2]. Although vaccines are one of the most significant achievements of public health, and their benefits outweigh their risks for the majority of the population, increasing hesitation and refusal of vaccination have been reported all over the world [3]. In 2019, the World Health Organization (WHO) declared vaccine hesitancy among the ten most influential global health threats [3]. During the COVID-19 pandemic, many healthcare workers found themselves unprepared to deal with user rejection and scepticism. Despite the high case fatality rate, the overwhelming overload of patients’ admissions to hospitals, and the efforts made by the public health authorities, vaccine hesitancy towards COVID-19 vaccines was frequently reported [4,5].

Numerous factors have influenced the level of trust and adherence of the population to the vaccine campaign and to other mitigation measures for COVID-19; in particular, the novelty of the disease, the unusually rapid vaccine development, the use of new vaccine technologies, and the rapid and uncontrolled spreading of information and misinformation (i.e., “infodemia”) were reported among the main factors [6]. Furthermore, the highly evolutive scenario and the rapid evolution of evidence obliged governments and institutions to recurrent changes of course and forced healthcare workers to daily adaptations, further alimenting confusion in the population [7].

During the COVID-19 vaccination campaign, healthcare workers, who represented the main point of contact for the population, faced a threefold challenge: adapting to government and institutional directives, justifying the uncertainty to users, and confronting a more vaccine-hesitant population. Evidence shows that frontline healthcare workers are among the most trusted sources of vaccination information and advice, and thus, one of the main factors in the success/failure of a vaccination program [8]. Multiple studies demonstrated that receiving pertinent information from healthcare workers helps hesitant users in developing vaccine awareness and increases vaccine acceptance [9,10,11].

Carefully planned and proactive communication is one of the principal tools at the disposal of healthcare workers to improve vaccine acceptance during vaccination consultation [12,13]; effective communication helps to hinder misinformation and anti-immunization messages [14]. The COVID-19 pandemic indicated that this tool is—more than ever—essential to creating demand for preventive services. Before the pandemic, research concerning health communication mainly focused on childhood vaccination or health communication in general [15,16,17,18]; only a few analyzed vaccination campaigns carried out on the adult population [19,20], and the case of health communication about COVID-19 vaccine is pretty unique in various perspectives, as mentioned above.

In order to deal with hesitancy and vaccination refusal, it is of pivotal importance—both for institutions and healthcare workers—to develop communication strategies based on the experiences of those who worked on the front line during the vaccination campaign. Indeed, vaccinators represent a privileged point of view that may allow us to capture the different nuances of vaccine attitudes of users throughout the vaccination campaign and identify the determinants associated with them. Furthermore, through their experience, it is possible to comprehend the communication needs of different categories of users and define which strategies were the most effective during the evolving epidemic context. Therefore, the present study aimed to characterize public attitudes toward COVID-19 vaccination faced by vaccinators and factors influencing vaccine confidence and identify communication strategies adopted during vaccination consultation by vaccinators.

## 2. Materials and Methods

The study was conducted according to the Helsinki principles, and informed consent was obtained from all subjects involved in the study.

### 2.1. Setting, Design and Population

Italy has been one of the first European countries to be severely hit by the COVID-19 pandemic [21]. To contain the spread of the virus, the Italian government established a national lockdown from 10 March to 4 May 2020 [22]. The subsequent months of 2020 were characterized by the gradual reopening of services, but physical distancing rules and face masks remained mandatory [23]. The vaccination campaign started on 27 December 2020 [24], and it was organized according to the *National Strategic Vaccine Plan for the Prevention of SARS-CoV-2 Infections*, which gave priority to the elderly and the frail [25]. Once more vaccine doses became available, the vaccination campaign gradually extended the age groups for which it was possible to get the vaccine; on 16 June 2021, the vaccination was opened to anyone over the age of 16.

This qualitative study was conducted on a sample of healthcare workers involved in the COVID-19 immunization campaign in the Tuscany Region (the Province of Florence, Province of Pisa, Province of Grosseto, and the Province of Pistoia); in particular, vaccinators from different services were recruited: mass-vaccination campaigns (HUB), general practices (in the first phase of the vaccination campaign, general practitioners (G.P.) vaccinated the frail and elderly, and subsequently, other users, too), hospital vaccination services (in charge of vaccinating healthcare workers and frail inpatients), and USCA services (*Unità Speciali di Continuità Assistenziale*—special continuity care units, a service in charge of the vaccination of frail people at home and in nursing homes, as well as of refugee and migrant populations).

Purposive sampling was used in order to select participants from the different services according to the following urbanization level areas: urban areas, suburban areas or rural areas. Participants were asked to check their records about COVID-19 vaccination; G.P. participants were also asked to check records about patients enrolled in their practice. Vaccinators who had performed at least 1000 vaccination consultations and general practitioners who had at least 750 patients registered in their practice were included. Furthermore, in order to cover distinct phases of the vaccination campaign and the COVID-19 epidemic, vaccinators who did not take part continuously (at least 12 months) in the vaccination campaign were excluded. The participant recruitment was stopped when thematic saturation was reached; this was discussed and agreed upon among the authors. Participants were contacted and interviewed between April and November 2022.

### 2.2. Data Collection, Processing and Analysis

All participants were informed about the purpose of the study, and informed consent was obtained from all of them prior to their participation in the study. Data were collected through in-depth face-to-face interviews jointly conducted by two researchers (G.A.; M.F.C.). No relationship was established prior to the study commencement between participants and interviewers. The research team developed a semi-structured interview to explore the following domains: users’ information-seeking behaviors, perception of COVID-19 vaccination, and communication strategies implemented during consultation. The interviews lasted for about one hour. In order to test the interview guide, a pilot test was conducted; interviewers were trained not to suggest value judgements.

All the interviews were audio recorded and transcribed verbatim; no notes were taken during the interviews by the interviewers. Participants were informed that data from interviews would be collected in an anonymous form. To examine the content of the corpus, an inductive thematic analysis was conducted. The first phase included repeated readings of the interviews to familiarize the interviewers with the data. Subsequently, a coding framework (codebook) was developed to organize the data-coding process. After the examination of the codes, the subsequent step involved the identification of the themes emerging from the data. Codes and themes were identified from the data by three authors (G.A.; M.F.C.; P.B.) independently and revised jointly. Disagreements on coding and themes were resolved involving a fourth author (V.L.) where necessary. The names of themes to be included in the final report have been reviewed by all the researchers. To facilitate data management and coding, the transcripts were uploaded and analysed with Iramuteq software (GNU General Public License version 2, GPLv2).

Furthermore, determinants of vaccine acceptance/hesitancy concerning COVID-19 vaccination that emerged from interviews were identified and traced back to the conceptual model developed by the WHO Strategic Advisory Group of Experts (SAGE) [16]. Specifically, this model identifies three macro groups of factors influencing the vaccine decision-making process: individual and group influences, contextual factors, and vaccine/vaccination-specific issues. The definition of determinants of vaccine hesitancy applied was “concepts relating to barriers and enablers for uptake, reasons for vaccination refusal, beliefs and attitudes towards vaccination and system mediated factors” [26].

## 3. Results

The interviews were submitted to a total of 30 vaccinators (46.7% female) aged between 25 and 71 years old. The characteristics of the vaccinators are summarized in Table 1. Appendix A reports the characteristics of the vaccinators at an individual level.

From the inductive thematic analysis of the interviews, four main themes emerged; below, the four identified themes are summarized.

### 3.1. Theme 1—Users’ Attitudes towards COVID-19 Vaccination

Interviews with vaccinators revealed six categories of users’ vaccine attitudes: eager to get vaccinated, compliant with the recommendations, information/reassurance seekers, wait-and-seers, unwillingly vaccinated to avoid work and social restrictions, and ideologically opposed to the vaccine.

The respondents reported that the preponderance of the population was eager to get vaccinated; this attitude was related to a generally high level of confidence in the health system, science and vaccine efficacy and was observed particularly in the elderly and young adults for different underlying reasons, i.e., perceived susceptibility to COVID-19 and regaining social life, respectively.

Users compliant with the recommendations were described as those who were vaccinated on time without any doubt or concern and who did not want any further information, mainly because they trusted science and were willing to pursue their duty as citizens.

Other users delayed vaccination for fear of the vaccine; two distinct categories of such hesitant users were described: reassurance seekers and wait-and-seers. The first group openly shared their fears during the consultation, while the wait-and-seers were those mainly observed in the early phases of the vaccination campaign who waited to see the effects of the vaccine on those who received it; a substantial distrust in science and institutions was at the basis of this attitude. Lastly, some users reported a high level of mistrust in the health system and institutions and believed in conspiracy theories. These users frequently did not trust the scientific community even before the pandemic; the rapidly evolving scientific evidence and contradictory directives for pandemic management exacerbated this attitude. Some of these users eventually became vaccinated to avoid work and social restrictions, while others firmly pursued their beliefs and never became vaccinated. Generally, users belonging to these two categories were middle-aged, had an average to good education level, and belonged to the middle socio-economic class. The most relevant quotations for this theme are summarized in Table 2.

### 3.2. Theme 2—Factors Influencing Vaccine Confidence

#### 3.2.1. Sub-Theme 2.1—Determinants of Vaccine Hesitancy: Individual and Group Influence; Contextual Influences; Vaccine/Vaccination Specific Issues

All the determinants of vaccine hesitancy and confidence that emerged from the interviews were categorized according to the SAGE Working Group model. Factors identified as either barriers or promoters of vaccine confidence are shown in Table 3.

The main barriers which lowered vaccine confidence were related to the rapid development of a new vaccine and the general belief that there were not enough clinical trials to guarantee vaccine safety. In particular, a widespread tendency among users has been to implement a heuristic risk-benefit calculation rather than one based on scientific evidence. Moreover, excessive exposure to mass-media information and a lack of clear guidance from institutions fostered users’ concerns (see theme 3).

COVID-19 normative/institutional restrictions, perceived disease severity, mode of delivery and accessibility of COVID-19 preventive services were identified as major promoters.

Lastly, socio-economic, religious, and cultural factors, as well as sex, did not emerge as significant determinants. The most relevant quotations for this subtheme are summarized in Table 4.

#### 3.2.2. Sub-Theme 2.2—Changes in User Vaccine Confidence over the Course of the Vaccination Campaign

Vaccinators reported an evolution of users’ vaccine confidence during the different pandemic phases. In the early stages of the pandemic, when the vaccine was not yet developed, the perception of the disease severity was very high, and people strongly desired the possibility of having an effective vaccine. This explains the positive attitude observed at the beginning of the vaccination campaign. Vaccine confidence increased even more during the first phases of the vaccination campaign as more users became vaccinated without relevant side effects reported. In the subsequent stages, vaccine confidence increase was hampered and probably reversed due to various reasons. The first was related to the report of some cases of severe vaccine side effects amplified by the mass media. The second was the presence of different types of vaccines, with some believed to be safer or more effective; this generated a widespread willingness to opt for the preferred one, even though it was not possible to choose among vaccines. The third is related to the reduced perception of COVID-19 disease severity when a high level of immunization coverage in the population was reached. The fourth is related to the misleading messages—both from mass media and institutions—regarding the fact that the vaccine could protect from contracting the infection led to an increase in vaccine hesitation when infections in vaccinated people were raised. Furthermore, pandemic fatigue led to a general demotivation toward recommended preventive measures, including vaccination. In the most recent stages of the vaccination campaign, with the almost complete removal of containment measures and the return to a pre-COVID-19 social life, vaccination was perceived as safe as other routine vaccinations. The most relevant quotations for this subtheme are summarized in Table 4.

### 3.3. Theme 3—Information and Health Communication

#### 3.3.1. Sub-Theme 3.1—COVID-19 Vaccine Knowledge and Information-Seeking Behaviors

Interviews highlighted that users generally utilized horizontal communication channels (i.e., mass media, social media, friends and relatives) and very few used vertical and official communication channels (i.e., responsible institutions). In this context, the main source of information for the elderly were mass media, general practitioners, and relatives, while the Internet and social media were mainly used by the young and middle-aged population groups; the population group aged 45–60 years was perceived more vulnerable to the Internet and social media misinformation. Some vaccinators observed that a frequent information-seeking behavior unconsciously applied by users was picking the sources that confirmed their presumptions and avoiding the ones that could refute or belittle their beliefs (i.e., “cherry picking”); this caused a bias and led an individual to validate the desired thesis, even when false. Interestingly, interviewees noticed that, although this phenomenon was common to most users, it was stronger in those who showed contrary or distorted ideas about vaccination.

Lastly, vaccinators from the HUB service highlighted that adherence to the vaccination campaign could have benefited from a more proactive contribution in terms of vaccine advocacy from the general practitioners, especially in people aged below 60 years. Indeed, it was reported that they would have benefited—both in terms of user vaccine confidence and time spent in consultation—by also having these users previously informed by the most proximal and trusted health information source, i.e., the general practitioner. The most relevant quotations for this subtheme are summarized in Table 5.

#### 3.3.2. Sub-Theme 3.2—The Impact of Institutional and Media Communication on Users’ Vaccine Perception and the Related Consequences on Vaccinators

Vaccinators perceived that users had access to an enormous quantity of information about the COVID-19 vaccine and that the vaccine topic became of public wisdom as never before. A discrete proportion of users were reported to be highly impacted by this unfiltered amount of information, misinformation, and disinformation; this fuelled uncertainty, a general sense of helplessness, and, consequently, fertile ground for vaccine hesitancy and conspiracy theories.

Furthermore, vaccinators reported a lack of guidance from institutions in terms of crisis communication to the population; users were confused by the inconsistency and the rapidly evolving institutional directives on the pandemic and vaccine campaign. This, together with the infodemia, further increased mistrust and vaccine hesitation and undermined the credibility not only of the institutions but also of the vaccinators, who have felt abandoned in the task of informing users. Vaccinators noticed that more consistent, transparent, and coordinated communication from the involved public institutions would have been fundamental to strengthening population trust and reducing the opportunity for rumours to spread, ultimately facilitating their work. Lastly, vaccinators highlighted the limits of the COVID-19 vaccine communication campaign that addressed the entire population indiscriminately, leaving a sense of vagueness. The most relevant quotations for this subtheme are summarized in Table 5.

### 3.4. Theme 4—Principles and Strategies for Dealing with Vaccine Hesitancy and Refusal

All the vaccinators reported that their professional opinion matters, and they felt influential in the user decision-making process. Several communication strategies for dealing with hesitant users emerged from the interviews; key principles and effective communication strategies for dealing with hesitant users were mainly inherent in the following areas:Empathy: it emerged how crucial it was to have active and non-judgmental listening, to make the user feel understood, and to ask for users’ feedback about the information provided during the consultation. Furthermore, putting themselves on the same level as the user was regarded as a useful strategy by several vaccinators, with some of them saying that sharing their own experience and initial concerns about the vaccine reassured hesitant users. Lastly, another strategy reported to reassure hesitant users was to involve other users’ trusted medical health professionals so as to make them feel truly heard and understood.Transparency: transparency and honesty about scientific evidence was deemed essential to establishing a relationship of trust with users. Furthermore, it highlighted the importance of clearly admitting the uncertainty of evidence when they were not strong enough. This situation occurred, for example, in the case of the vaccination of pregnant women at the beginning of the vaccination campaign.Highlighting the balance between risks and benefits: vaccinators found it useful to stress the high level of vaccine efficacy and safety on the one hand while highlighting the risk of getting COVID-19 on the other hand. The use of everyday life examples, such as pointing out that the COVID-19 vaccine had similar or fewer risks than other commonly used drugs, was useful in overcoming fear and hesitation. Since one of the most common fears was the unknown long-term effects of the vaccine, pointing out that COVID-19 also may have long-term effects was proven useful. Furthermore, highlighting specific aspects in terms of personal or familiar clinical risks helped to make users understand that vaccination is particularly useful for their case.Adopting a personalized approach: a user-centered approach was frequently mentioned in dealing with vaccine hesitancy. This required exploring reasons behind hesitancy without being judgmental and understanding the user’s belief and attitude toward health and vaccines in order to adapt the communication style. If the user shows beliefs and values in line with vaccine recommendations, reporting scientific data and evidence was reported to be persuasive; in the case of contrasting convictions, emotional stories were reported to be more effective.

Lastly, many vaccinators pointed out that explaining the importance of vaccination in order to achieve community immunity (i.e., collective responsibility) did not influence the vaccination choices; moreover, all the vaccinators did not use or share communication-supportive materials during the consultation since they noticed that users preferred to hear information directly from them.

As for users ideologically opposed to the vaccine, two different cases of dealing with this category emerged: the case in which they were actively called by their physician to receive the vaccination and the case in which they spontaneously came up to the vaccination HUB (usually merely to protest). In both cases, they were not open to discussion, hardly or never deviated from their beliefs, and sometimes verbally harassed vaccinators. Apparently, there is no effective communication strategy for developing an informed decision-making partnership. Interestingly, in the case of the general practitioners, the divergent opinions on the vaccine with users ideologically opposed did not hamper the patient-physician relationship in other health areas as mainly they believed that the G.P. themself was a victim of a conspiracy from institutions/pharmaceutical companies. Given the high levels of tension involved in the conversations with this type of user, avoiding dismissing and debating and instead providing clear and concise recommendations was deemed essential as a first approach to building trust; this may help in creating further opportunities to discuss the vaccination more openly. The most relevant quotations for this theme are summarized in Table 6.

## 4. Discussion

Several studies demonstrated that receiving pertinent information from health operators may increase vaccine acceptance [8,9,10,11]. Through the experiences of vaccinators, this study aimed to characterize public attitudes toward COVID-19 vaccination and the communication strategies adopted during vaccination consultations. The study was based on face-to-face in-depth interviews with vaccinators who had continuously worked during the different stages of the COVID-19 vaccination campaign for at least 12 months so as to draw evidence from a privileged viewpoint on the attitudes and beliefs of the population and on effective strategies for vaccine communication with users. The content analysis of the interviews highlighted four themes; the first and second themes described users’ attitudes towards COVID-19 vaccination and barriers or promoters influencing their decision-making about the vaccine. The third theme highlighted the information-seeking behaviors of users and the impact that the “infodemia” had on user vaccine confidence; and lastly, in the fourth theme, communication principles and strategies for dealing with vaccine hesitancy and refusal emerged.

The interviews revealed several profiles of users’ attitudes toward COVID-19 vaccination. The profiles described reflect the vaccine hesitancy continuum model described for routine vaccinations [13], with some specificities related to the case of the COVID-19 vaccine. Indeed, the rapid development of the vaccine and the new vaccine technology are probably the main underlying reasons behind the wait-and-see attitude observed in some users, while the stringent work and social restrictions foreseen for the unvaccinated had forced some of the more reluctant users to put aside their beliefs and become vaccinated. Furthermore, the mass vaccination campaign and the high media coverage had probably pushed, on the one hand, more people to get vaccinated, while on the other hand, it increased hesitancy in some users due to the infodemia.

Interestingly, all the vaccinators described a similar trend of vaccine confidence over the course of the vaccination campaign (depicted in Figure 1); vaccine confidence was reported to be high at the start of the campaign and to increase during the first phases—in this context, it should be taken into account that Italy was one the first western countries to be severely hit by the spread of the pandemic [22,27,28]; then, this trend stopped and reversed for several reasons, some directly related to the advancement of the vaccine campaign (the appearance of the first vaccine adverse events and the reduced perception of the disease severity due to the high level of vaccine coverage) and other external factors (i.e., misinformation/disinformation and pandemic fatigue) [29]. With the return to a pre-COVID-19 situation, the COVID-19 vaccine begins to be internalized by the population as a routine vaccination in terms of safety. These trends are of particular interest and should be taken into account, as they will presumably replicate in the event of future pandemics.

Concerns over vaccine safety and fear of unknown long-term health consequences were among the main factors observed behind vaccine hesitancy for the COVID-19 vaccine. These factors were mainly related to the rapid development of the vaccine and to the novelty of the mRNA technology, as the main belief was that the vaccine had not been tested enough. These findings are in line with other studies carried out in different countries and settings [30,31,32,33,34]. Furthermore, socioeconomic, cultural factors and sex did not emerge as relevant determinants in our study; interestingly, studies from other high-income countries have observed a higher COVID-19 vaccine hesitancy in females, in people with lower levels of education, and in people not being of white ethnicity [30,35,36,37].

Findings also showed that these concerns and beliefs were highly exacerbated by the lack of clear guidance and communication from the government and competent health institutions and by the infodemia. Indeed, the experience of vaccinators highlighted that the COVID-19 pandemic had the characteristics—such as the novelty of the disease, the evolving situation, the need for rapid adaptations of policies and public health measures—to create fertile ground for the spread of rumour and that users were not able to navigate the huge amount of available information. Misinformation and disinformation created fears and discouraged people from their decision to get the vaccine [38,39,40]. Although the impact of fake news on vaccination campaigns has been widely observed in the context of other vaccines [41,42], during the pandemic, the viral spread of doubts travelled at unprecedented speed, especially on social media [43]. A more consistent and coordinated vaccine communication from the involved public institutions appeared to be fundamental to counter misinformation and to strengthen population trust. As seen in this pandemic and in other previous pandemics, feeling well-informed about the pandemic and establishing a relationship of trust between public institutions and the population seems to play a critical role in order to engage the population in vaccination campaigns [44,45,46,47].

During consultations with patients, vaccinators highlighted that their opinion matters and felt they were a trusted source of information for hesitant users. This finding confirms that first-line healthcare workers have a prominent role in guiding hesitant users and in increasing vaccine acceptance [8,9,10,11]. It has to be taken into account that healthcare workers in Italy had a positive public image during the pandemic, given their efforts during the health crisis and their great level of COVID-19 vaccination uptake—higher than in other high-income countries; this could have increased their credibility among the users [37,48,49,50]. Empathy, bringing first-hand and personal examples, exploring and legitimising patients’ fears (rather than dismissing them), and ensuring transparent communication appear to be effective in building trust and overcoming vaccine hesitancy. Moreover, a personalized approach and a tailored communication style according to the user attitudes and beliefs were other key principles that emerged in dealing with vaccine-hesitant users. These aspects are particularly important as risk perception—of both the disease and the vaccine—and vaccine hesitancy are multifaceted phenomena that are influenced by personal experiences and socio-cultural and economic backgrounds [51,52,53].

Interestingly, our findings clearly characterize a subgroup of users that were ideologically opposed to the vaccine; in line with other studies, our findings showed that these users had an a priori refusal of the vaccines, a high level of distrust in conventional medicine, health system and institutions, and a tendency to believe in conspiracy theories [54,55]. In this group of users, it seems that there was no room for discussing vaccination, sharing knowledge of the vaccine, and developing an informed decision-making partnership. Avoiding debating evidence, providing brief and clear recommendations, and being available for further consultations may be of help in establishing further occasions for discussing vaccination with this kind of user.

The present study had several strengths and limitations. As for the strengths, the maximum variation purposive sampling allowed to include the widest range of perspectives possible—i.e., vaccinators working with different types of users and in distinct services based in urban and rural contexts—so as to explore the topic from different angles and identify both common patterns and variations. Furthermore, the study included vaccinators that continuously vaccinated for at least one year; this allowed us to have a solid overall perspective on the evolution of vaccine confidence over time and on the challenges faced in the various vaccination campaign phases. As far as the limitations are concerned, the sample was selected from only one Italian region. Furthermore, as the study is based on data from face-to-face interviews, a social desirability tendency of participants cannot be dismissed; however, interview questions were specifically developed in order to be neutral, and interviewers were trained not to suggest value judgements or interpretations.

## 5. Conclusions

Our study explored the privileged viewpoint of vaccinators on the COVID-19 vaccine campaign. Our findings highlighted aspects and trends of user vaccine confidence that could recur in cases of future pandemics or rapid vaccine development, especially in the context of mass-vaccination campaigns and high public and media attention. At the population level, consistent, targeted and coordinated vaccine institutional communication is needed in order to build trust with the population and to empower the frontline healthcare workers involved in the vaccination campaign. At the individual level, our findings confirm that frontline operators are essential in dealing with vaccine hesitancy and in increasing vaccine acceptance. Several key communication principles and strategies were identified to foster vaccine confidence; while, although it may be proven difficult, avoiding debating and providing brief and clear recommendations may serve to start building trust with users ideologically opposed to the vaccine and to leave the door open for further occasions. Further studies that include user perspectives are needed in order to better identify leverages and strategies to engage with this population subgroup.

## Figures and Tables

**Figure 1 vaccines-11-00223-f001:**
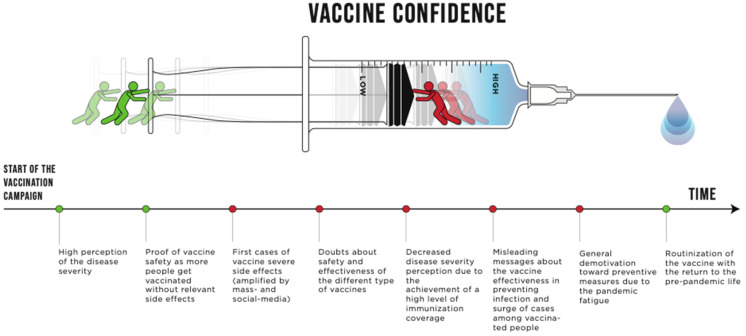
Evolution of vaccine confidence over the course of the COVID-19 vaccination campaign.

**Table 1 vaccines-11-00223-t001:** Study sample characteristics.

Total Sample *n*	30
**Median age** (interquartile range)	37 (31–44)
**Female sex** *n* (%)	14 (46.7)
**Number of vaccinations administered** *n* (%)	
1000–2000	17 (56.7)
2000–5000	6 (20)
5000–10,000	4 (13.3)
>10,000	3 (10)
**Number of patients** * (mean ± standard deviation)	1405 ± 253
**Previous experience with other vaccination campaigns** *n* (%)	19 (63.3)
**Urbanization level area** *n* (%)	
Urban	10 (33.3)
Suburban/rural	20 (66.7)
**Type of vaccinators** ** *n* (%)	
Vaccination HUB	12 (40)
Hospital	3 (10)
Special continuity care units	6 (20)
General practitioners	12 (40)

* Only for general practitioners; ** The total sum is higher than 100% as some vaccinators worked in more than one vaccination service.

**Table 2 vaccines-11-00223-t002:** Illustrative quotes for theme 1, “users’ attitudes towards COVID-19 vaccination”.

Theme 1. Users’ Attitudes towards COVID-19 Vaccination
- There are several type of user attitudes: those who are enthusiastic about the vaccine, they want to be vaccinated immediately, and they would take also an extra dose, if it were possible; then there are the passive users, who have been told that “this is the rule and it has to be done” so they do it without expressing any particular opinion about vaccine, if it has to be done it has to be done; then there are the reassurance seekers, those who are afraid that the vaccine will hurt them, but in the end they do get vaccinated; there are the opposers, who don’t want to get it for ethical or political reasons...basically, these are the categories (Vaccinator n. 18, G.P.). - (…) the majority of people trusted the doctors, the health system and the safety of the vaccine. (Vaccinator n. 3, HUB). - (People) came confident and happy to have this chance (to get vaccinated) in order to come back to social daily life in the most normal conditions as possible. For most people, the goal was this (to return to normal life) (Vaccinator n. 11, HUB). - (...) youngsters, like those in their twenties, were receptive and willing to do it. Teenagers are the ones who have made a very rational thought: “I want to get back to my life, that’s what the vaccine is for, so I am gonna do it” (Vaccinator n. 18, G.P.). - (…) some people just follow the rules as they were dictated to them; those subjects act without making any criticisms or assessment: if the rule says a thing has to be done, they just make it (Vaccinator n. 17, G.P.). - They are not contrary to the vaccine because of the information they had read on the Internet. The hesitant people are looking for answers, so if you give them the right amount of time and attention, they will eventually become users convinced of what they are doing; it takes time, but the hesitant people don’t want per se to do it...they just need an extra communicative effort (Vaccinator n. 10, HUB). - (…) there were users that were afraid; I mean, they did not distrust the health system or the health worker, but they had a lot of apprehension and doubts generated by the news (Vaccinator n. 11, HUB). - Some waited because they were afraid, they said “now let’s see how many people get vaccinated, and let’s see if they will have any issues” (Vaccinator n. 23, G.P.). - Covid-19 vaccines have been administered in a short time frame, so (they say) “Alt! before doing it myself, I will think about it”; it’s a very selfish kind of argument based on the conviction that other vaccines have been tested on people, whereas Covid-19 vaccine was not, so (they said) “why should I be the guinea pig?” (Vaccinator n. 15, HUB). - There were people who came for vaccination and told me “I only get vaccinated because I have to, otherwise I can’t work” (…) these patients were the most difficult to handle (Vaccinator n. 14, USCA). - Telling them (users ideologically opposed to the vaccine) that vaccination is beneficial for their health and can save their lives it’s not compelling; the only way is to affirm that there are norms that require the vaccination, and those laws must be obeyed; they may get vaccinated only because of that (Vaccinator n. 6, HUB). - This small percentage of users (ideologically opposed to the vaccine) is firmly convinced.. it will be almost impossible to change their minds; they are probably the ones who are still not vaccinated (Vaccinator n. 1, HUB). - The “no-vax” people wouldn’t accept any form of dialogue: they yell at you, tell you to f**k yourself, and say you bye-bye; in my opinion, there are limited activities that can be done with them (Vaccinator n. 3, HUB). - They are often people with an average level of education—neither illiterate nor graduates... they are neither very poor nor very rich, so let’s say (that they belong to) the middle class of the population (Vaccinator n. 18, G.P.). - The “no-vax” people were users with a medium-high cultural level—usually not in the medical field; they presume to have all the answers without any needing to consult you (Vaccinator n. 18, G.P.).

**Table 3 vaccines-11-00223-t003:** Factors identified in the interviews as either barriers (B) to or promoters (P) of COVID-19 vaccination and mapped onto the SAGE Working Group model of determinants of vaccine hesitancy.

		B	P	BARRIERS	PROMOTERS
		N. vaccinators			
**SOCIO ECONOMIC GROUP**	**Age**	22	21	- 45–60 years	- Over 70 years - Under 30 years
	**Race/ethnicity**	3	-	- Residents and migrants from East Europe	
	**Birthplace**	3	-	- Hard-to-reach migrant populations	
	**Income/SES**	1	5	- Lower-middle class	- Precarious working conditions - Economic migrant
	**Marital status/family composition**	-	-		
	**Education**	8	1	- Average to good educational background/level (excluded education in the health science fields)	- Low education level
	**Occupation**	-	2		- Healthcare workers
	**Language proficiency**	-	-		
	**Family decision making**	4	2	- Having a family member ideologically opposed to the vaccination (in the case of elderly and adolescent vaccination)	- Having a family member vaccine confident (in the case of elderly and adolescent vaccination)
	**Access to health care**	-	-		
	**Health status**	9	2	- Pregnancy - Allergies and taking pharmaceuticals for other illnesses - Comorbidities (cardiologic, rare and autoimmune diseases) - Frail users	- Frail users - People who live in nursing home
	**Family size**	-	-		
**RELIGION/CULTURE/GENDER**	**Religious affiliation**	-	-		
	**Cultural**	2		- Ethical reasons	
	**Gender**	-	-		
**POLITICS/POLICIES**	**Politics**	3	-	- Government distrust - Supporting political parties that are against policies promoting vaccination and containment measures	
	**Policies**	8	14	- Conflicting directives - Work and social restrictions foreseen for the unvaccinated	- Work and social restrictions foreseen for the unvaccinated
**INFLUENTIAL LEADERS AND INDIVIDUALS**	**Influential leaders and individuals**	1	-	- Influential religious leaders followed by some migrant communities	
**COMMUNICATION AND MEDIA ENVIRONMENT**	**Access to information**	-	5		- Easy access to information from mass-media - Institutional communication campaigns
	**Mass Media (use and influence)**	19	-	- Infodemia - Fake news - Misinformation/disinformation from social/mass-media - Sensationalistic type of journalism - Institutional communication campaigns	
**PHARMACEUTICAL INDUSTRY**	**Pharmaceutical Industry**	11	-	- Conspiracy theory	
**HISTORICAL INFLUENCES**	**Historical influences**	-	-		
**GEOGRAPHIC BARRIERS**	**Place of residence**	-	-		
**RISK/BENEFIT (SCIENTIFICALLY BASED)**	**Use of evidence**	-	4		- Seeking medical evidence from official sources
	**Trust in evidence**	2	1	- Distrust in the scientific process	- Confidence in science
**VACCINATION SCHEDULE**	**Schedule**	2	-	- Unclear schedule for those who had previously experienced COVID-19 (early stages of the campaign) - Not having clear whether further booster doses would be needed in the future	
**MODE OF** **ADMINISTRATION**	**Mode of administration**	2	-	- Agophobia	
**MODE OF DELIVERY**	**Campaign**	2	3	- Rigid rules for appointment scheduling in the early stages of the vaccination campaign - The possibility to book the vaccination appointment only online in the early stages of the vaccination campaign	- Easy rescheduling of vaccination appointment - Open access and evening opening of the vaccination HUBs - Cooperation between primary care services for the vaccination of the hard-to-reach populations
**INTRODUCTION OF A NEW VACCINE OR NEW FORMULATION**	**Introduction of a new vaccine or a new formulation**	17	-	- New technology in the vaccine - Rapid development of the vaccine - Absence of real world-data on vaccine side-effects	
**RELIABILITY OF VACCINE SUPPLY**	**Supply**	-	-		
**ROLE OF HEALTHCARE PROFESSIONALS**	**Patient communication**	-	16		- Involving other trusted health professionals in the decision-making process - Transparent and honest communication of evidence - Empathy and active and non-judgmental listening - Sharing own experience and real stories about the vaccination
	**Vaccination expectations**	10	-	- The unmet expectation that the vaccine would prevent SARS-CoV-2 infection	
	**Organizational culture**	-	-		
	**Place of work**	-	-		
**COST**	**Financial**	-	-		
	**Time**	-	-		
	**Administrative**	-	-		
	**Access**	-	-		
**TAILORING VACCINES/VACCINATION TO NEEDS**	**Options**	11	-	- Concerns about the possibility of using different types of vaccines	
**EXPERIENCE WITH PAST VACCINATION**	**Vaccination behavior**	2	1	- Having refused other adult vaccinations	- Having received the flu vaccine
**RISK/BENEFITS (PERCEIVED /HEURISTICS)**	**Susceptibility to disease**	3	4	- Not believing in being at risk of contracting the infection	- Fear of the infection - COVID-19 personal experiences (or in the closest people)
	**Disease severity**	3	11	- Not believing in being at risk of severe COVID-19 - Fear of the vaccine greater than fear of the disease	- High perception of the risk of the disease - Dramatic memories of COVID-19 deaths observed in 2020 - Severe COVID-19 personal experiences (or in the closest people) - Perception of the severity of the disease in the elderly and in other population subgroups at risk
	**Vaccine safety**	25	4	- Negative experience with previous doses - Fear of long-term reactions (e.g., genome modifications, cancer, fertility, etc.) - Fear of side-effects - Concerns about the composition of the vaccine (mRNA) - Safety in pregnancy and breast-feeding	- Vaccine confidence attitude
	**Vaccine efficacy**	4	9	- Lack of protection against the infection	- Awareness of vaccine efficacy in preventing severe COVID-19
**PERSONAL EXPERIENCE WITH AND TRUST IN HEALTH SYSTEM AND PROVIDER**	**Distrust/fear of vaccine due to**	7	-	- Distrust in the healthcare system - Distrust of the G.P.s’ ability to identify a conspiracy	
	**Satisfaction with public health system**	-	4		- Trust and reliance on the doctor’s recommendation - Confidence in the healthcare system
**KNOWLEDGE/AWARENESS OF WHY/WHERE /WHAT/WHEN VACCINES ARE NEDEED**	**Knowledge—Vaccination**	2	2	- Lack of awareness of the steps and requirements necessary to have a commercially approved vaccine - Lack of vaccine literacy skills	- Parents well informed about their children’s vaccination - Being aware of vaccine’s benefits
	**Knowledge—General Health**	2	-	- Low health literacy level	
**BELIEFS, ATTITUDES AND MOTIVATION ABOUT HEALTH AND PREVENTION**	**Attitude**	4	2	- Risk-taking inclination - Ideologically opposed to conventional medicine	- Trust in science and in the scientific method
	**Beliefs**	4	-	- Belief in complementary and alternative therapies - Belief that drugs and conventional medicine are harmful	
	**Motivation/Practices**	1	-	- Use of alternative therapies for preventing/treating COVID-19	
**IMMUNISATION IS A SOCIAL NORM VS. IMMUNIZATION IS NOT NEEDED/** **HARMFUL**	**Need for vaccine**	2	2	- Considering the vaccine unnecessary	- Being aware that the vaccine protects one’s own inner circle - Awareness of community immunity mechanism and why it is needed to end the pandemic crisis - Attitude to respect the recommendation from the institution

**Table 4 vaccines-11-00223-t004:** Illustrative quotes for theme 2, “factors influencing vaccine confidence”.

Theme 2. Factors Influencing Vaccine Confidence
**2.1.—Determinants of Vaccine Hesitancy: Individual and Group Influence; Contextual Influences; Vaccine/Vaccination Specific Issues**
- Long-term fears were due to the fact that the vaccine was new and the belief that it was not tested enough (people said) “I am still fertile, who knows what kind of children I may have”, “I am worried about the possible changes in my DNA”, “who knows what will happen in thirty years when my children will be grown up”(Vaccinator n. 14, USCA). - It was a new vaccine, so people were really, really worried (Vaccinator n. 10, HUB). - I remember people saying “Uhm, but the vaccine is still experimental” (Vaccinator n. 11, HUB). - Some people asked themselves “What’s the point of getting vaccinated if I will get COVID-19 anyway” (Vaccinator n. 9, Hospital and HUB). - In my opinion, there has been an excess of information, which created confusion among people. From an institutional and a regulatory point of view, they have not been capable of giving the required few clear rules…also from a media point of view, the virologists in TV shows who said everything and more…for example, until the day before, it was said that the surgical mask is effective and then, the day after, it was no longer protective, many people thought “how is it possible that on 23 July the surgical mask is fine and on the 24th it is not adequate anymore? There must be something wrong here”; and this mechanism arouse suspicions, especially in people with a lower level of trust (Vaccinator n. 17, G.P.). - News about vaccines were continually and literally thrown at people, but they were not able to digest them; too many voices wanted to take part in this discussion; all this led to debates that fuelled insecurity in people’s minds (Vaccinator n. 1, HUB). - Some users said “I only get vaccinated because I have to, otherwise I can’t go to work” (Vaccinator n. 15, HUB). - At the beginning (of the vaccination campaign), we had a great adherence and the users we vaccinated were eager to get the vaccine, they had no hesitation... they perceived a high risk of COVID-19 (Vaccinator n. 4, HUB). - The vaccination campaign was well organized as it attempted to reach as many people as possible by opening the HUBs in the evening and allowing users to get vaccinated without prior booking...all this worked well (Vaccinator n. 11, HUB).
**2.2.—** **Changes in user vaccine confidence over the course of the vaccination campaign**
- (Vaccine confidence) changed in relation to COVID-19 risk perception: when the risk perception was high, vaccination was almost seen as it was the only way of salvation..everyone competed to get vaccinated first. Then, as time went by and risk perception declined, the vaccine almost became a hassle (Vaccinator n. 2, HUB). - (Vaccine confidence) has changed over the time...as the vaccine was given to more and more people, people realized that the vaccine doesn’t cause side effects and prevents people from getting sick or getting the severe forms (Vaccinator n. 3, HUB). - As the time went by and we went on with the third and the fourth doses, the proportion of sceptics increased..it seems that users had a higher perception of the risks of the vaccine rather than of those related to the disease...users were more hesitant (Vaccinator n. 4, HUB). - The negative news were amplified by the press; for example, the case of the side effects of AstraZeneca vaccine...this created huge concern in users (Vaccinator n. 8, HUB). - (…) a man, who previously received two doses of Pfizer vaccine, refused to receive the Moderna vaccine for the 3rd dose because he was afraid of possible side effects, he insisted to get the Pfizer vaccine (Vaccinator n. 5, HUB). - People were not well informed about the reasons why get the vaccine, it was not clearly explained the difference between preventing the infection and preventing the severe consequences...especially during the omicron wave it has been observed that (the vaccine) does not prevent the infection (Vaccinator n. 9, Hospital and HUB).

**Table 5 vaccines-11-00223-t005:** Illustrative quotes for theme 3, “information and health communication”.

Theme 3. Information and Health Communication
**3.1.—COVID-19 Vaccine Knowledge and Information Seeking Behaviors**
- The main sources of information were social media and television (Vaccinator n. 8, HUB). - In my opinion, TV was the main source of information…surely this medium reached the majority of people as it was switched on every day, at any time; moreover, it reached also those who did not use the web. Other sources of information were: the web, newspapers, radio, and at the very end the direct contact with the family doctor (Vaccinator n. 2, HUB). - Without any doubts, users mostly get their information from unofficial sources (Vaccinator n. 1, HUB). - Mostly they get information from Internet and television; I believe that older people and foreigners also get information from relatives and friends (Vaccinator n. 5, HUB). - (people seek information) only on Google…older people get informed from the TV. Youngsters, but also the not-so-young, “googled” everything…with “Google” I mean the use of Internet in general, including social-media. Government and institutions should push more and more on establishing a relationship (with users) through social media (Vaccinator n. 12, USCA). - (…) in the end, people look for what they are interested in and the answers they want to hear (Vaccinator n. 15, HUB). - The “no-vax” people seek information on the Internet by searching for facts they like to read…we all like to read things that confirm our own thoughts (Vaccinator n. 6, HUB). - The “no-vaxxers” are hyper-informed by sources that are selected by them...and they know very well that (these sources) are unreliable; they speak their own language, they “ring their bells”...they like that sound, they don’t want to hear another one (Vaccinator n. 12, USCA). - The weakest element has been the lack of integration with general practitioners and primary care services; the general practitioners represent the capillarity of the health system, so as to reach individuals with a trusted face (Vaccinator n. 1, HUB).
**3.2.—The Impact of Institutional and Media Communication on Users’ Vaccine Perception and the Related Consequences on Vaccinators**
- The several fake news spread on Telegram and Facebook were terrible. Especially users in their fifties were the more vulnerable to fake news, they were not able to discern a trusted source of information, and they believe in everything they read…they were those who mainly believe in conspiracy theories (Vaccinator n. 9, Hospital and HUB). - We’ve seen a huge amount of information, an infodemia. People’ve been intoxicated with information (Vaccinators n. 1, HUB). - (…) patients told me “we no longer know what to believe anymore”, there are too much information, both good and bad, obviously the bad is more problematic, but the good is also excessive; many people told me “I just don’t listen to the news anymore because it makes me anxious and it’s not clear what is necessary to do”, this has certainly created a lot of confusion in people (Vaccinator n. 19, G.P. and USCA). - During the vaccination campaign, everybody talked about that, everybody had to have an opinion about the vaccine and had to talk about it…more or less everybody had an idea, whether this idea was correct or not (Vaccinator n. 10, HUB). - Contrasting institutional messages generates confusion in users as well as in the health operators who had to deal with them and explain the reasons of the frequent change of directions…the users may think “you all are crazy!” (Vaccinator n. 11, HUB). - Most of the people don’t understand the scientific process and institutions were not able to make people understand why it was necessary to backtrack on some of the decisions made (Vaccinator n. 2, HUB). - The awareness-raising campaign was not done on a specific target group but on the entire population..it is difficult to involve all the age groups at the same time, they are receptive to different communication mechanisms (Vaccinator n. 1, HUB).

**Table 6 vaccines-11-00223-t006:** Illustrative quotes for theme 4, “principles and strategies for dealing with vaccine hesitancy and refusal”.

Theme 4—Principles and Strategies for Dealing with Vaccine Hesitancy and Refusal
- (with hesitant user) I try to establish a relationship of trust that can make them feeling safe (Vaccinator n. 2, HUB). - I tried to empathize with their fears, telling them that I have been afraid too, and that fear is a human reaction that must not be criticized but accepted (…) I mainly try to establish a close relationship (with the users) putting myself on the same level and provide accurate and transparent information whenever they had doubts (Vaccinator n. 1, HUB). - I told them (to the users): “I got vaccinated in January, I actually was among the first operators getting vaccinated, and thus I certainly had some fears and apprehensions, as it happens every time you try something new, but there were solid data (about COVID-19 vaccination) reliability and security. Even though I had some fear of getting vaccinated, I certainly had a greater fear of getting the disease”. This was something everyone agreed on (Vaccinator n. 3, HUB). - I tried to understand their main doubts and concerns, I tried to be as clear and understandable as possible, and surely I asked their feedbacks…I mean, when I was explaining something I checked that everything was comprehended. I used simple examples to describe the virus structure and how the vaccine works...I tried to simplify as much as possible (Vaccinator n. 10, HUB). - It is important to make them (hesitant users) understand that things are transparent, that the information is open for everyone, and that there are no hidden facts (Vaccinator n. 4, HUB). - In the case when I actually had no certainties and therefore I could not give any certainties (to the users), I spoke with honesty about the situation with complete honesty…this was helpful (Vaccinator n. 21, G.P.). - I always try to compare risks and benefits, let’s say they are concerned about the long-term consequences of the vaccine, I explained that there is no scientific evidence to fear the long-term consequences of the vaccine, whereas long-term consequences from the virus have to be feared (Vaccinator n. 2, HUB). - (I focused) on the benefits of vaccination compared with its side effects (such as fever), such as how important the vaccination is to protect from severe forms of the disease; I tried to make people understand that getting COVID-19 could result in severe consequences for their health (Vaccinator n. 15, HUB). - We try to make people understand that it is a balance between the expected benefit of protection and the consequences of the disease, including the long-term ones...stressing the fact that we don’t know exactly the long term consequences of the disease (…) there were always concerns about the long-term consequences of the vaccine, but we still don’t know what the long-term consequences of COVID-19 are (Vaccinator n. 11, HUB). - Sometimes an effective strategy for convincing a person to go ahead [with vaccination] is to involve other trusted professionals (…) there have been times when I have contacted their doctors just to give them a reassurance (Vaccinator n. 1, HUB). - In some cases I showed and compared the aspirin leaflet with that of the vaccine (Vaccinator n. 6, HUB). - Comparisons with other drugs considered harmless were also useful, highlighting the fact that in case they had taken these drugs in the past they had been exposed to similar risks of the vaccination (Vaccinator n. 21, G.P.). - With hesitant users the counselling was quite demanding, I assessed their risk both in terms of social interaction and clinical history, trying to highlight their personal risk. I remember a case of a hesitant couple, where one of the two had more indications to get the vaccine (…) in this case I remember telling to the husband “let’s say your wife might even wait for the moment as she’s a housewife and almost never leaves the house, but you have heart issues, you are diabetic, you work in a school, your situation cannot wait”...it is very useful to make them understand their own personal risks…I really have a personalised approach to the vaccine consultation (Vaccinator n. 19, G.P. and USCA). - In my opinion, people mostly don’t care about collective health, they mainly look at their own, they don’t care if vaccination reduces virus transmission, they just care about their well-being. From this point of view, the average person who comes to get vaccinated is very selfish (Vaccinator n. 9, Hospital and HUB). - (referring to the use of communication support materials) during consultation is much more important the face-to-face dialogue, people need to feel that they are talking to a reliable person who gives them a sense of security (Vaccinator n. 2, HUB). - (referring to the users ideologically opposed to the vaccine) there are no effective approaches, it’s like playing chess with a pigeon, they don’t care, you can tell them anything and they won’t change their opinion. It is a waste of breath (…) there is nothing that can convince them (Vaccinator n. 9, Hospital and HUB). - (referring to the users ideologically opposed to the vaccine) I tried as I did with the other users (…) but for them it is as if the earth was flat (Vaccinator n. 2, HUB). - They were convinced that on this particular issue (the vaccine) we (the doctors) have been fooled...in the sense that we were deceived too so we were doing things in good faiths. According to one of these theories, I was also manipulated, so they were not angry with me (Vaccinator n. 18, G.P.). - In my experience, there are no proper communicative approaches with the “no-vax” users…sometimes it seems they just want to provoke or to threaten us. Therefore, I think the best way for dealing with such users is avoiding any dismissal or direct confrontation, while at the same time try to provide concise recommendations...hoping that this may build some sort of trust for future occasions (Vaccinator n. 11, HUB).

## Data Availability

The data presented in this study are available on request from the corresponding author. Data are not publicly available due to the current regulation on privacy.

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
