# Peer review of "Communicating with Patients about COVID-19 Vaccination: A Qualitative Study on Vaccinators in Tuscany Region, Italy"

_vaccines, 2023, doi:10.3390/vaccines11020223_

Round 1

Reviewer 1 Report

Dear Author,

I suggest to remove to the “Theme 1 - Users’ attitudes towards COVID-19 vaccination” the ideologically opposed to the vaccine (line 156) since this group still get vaccinated to avoid work and social restrictions. Therefore this group coincides with the group of vaccinated to avoid work and social restrictions.

Author Response

We thank the reviewer for this comment, although users of these groups share the same beliefs (e.g. mistrust in the health system and institutions), the two groups are distinct as the ideologically opposed to the vaccine never get vaccinated while the other eventually get vaccinated in order to avoid work or social restrictions. We have better clarified this aspect in the result section of the revised manuscript (please see p. 4 lines 200, 205-206 of the revised manuscript).

Reviewer 2 Report

Dear editor,

Thank you for the kind invitation to review this manuscript. Overall the article is well written. Attached are my comments for the authors' consideration

The introduction is generally well written

Methods

- More information needs to be provided about the management of COVID-19 in Tuscany region

- It will be helpful to include any demographics related inclusion criteria and exclusio criteria.

- Would suggest to use the COREQ checklist to report the study

Results

- Given the small sample size, would suggest to report age as median and interquartile range. 

-> How do the participants know how many vaccination they have administered?

-> Table 1: Suggest to include all abbreviations used in the table as a footnote

-> Are there any information on demographics of the participants?

-> What does normative adherent mean?

Discussion

- It would be helpful to discuss any unique findings relative to existing studies done in the country vs neighouring countries and any novel implications if any. 

Minor comments

A brief word related to the general high rates of COVID-19 vaccine hesitancy and especially among healthcare workers should be highlighted in the introduction. 

Cite the following articles

https://pubmed.ncbi.nlm.nih.gov/35455286/

https://pubmed.ncbi.nlm.nih.gov/34452026/#:~:text=The%20rates%20of%20vaccine%20hesitancy,associated%20with%20increased%20vaccine%20hesitancy.

Author Response

Methods

- More information needs to be provided about the management of COVID-19 in Tuscany region.

We thank the reviewer for this comment, we have added more information about the management of COVID-19 in Italy and in Tuscany region in the materials and methods section. Please see p. 2-3 line 92-102 of the revised manuscript.

- It will be helpful to include any demographics related inclusion criteria and exclusio criteria.

We thank the reviewer for this comment. Please note that the inclusion and exclusion criteria of the study were not based on participants’ demographics, the study included healthcare workers involved in the COVID-19 vaccination services and general practitioners. We have provided further detail on participants’ demographics in table 1 and table S1 of the revised manuscript in order to better characterize the study sample.

- Would suggest to use the COREQ checklist to report the study

We thank the reviewer for this suggestion, we have used the COREQ checklist to report our study and have added the missing information where necessary. Please see p. 3 lines 121, 128-129, 132-135 of the revised manuscript.

Results

- Given the small sample size, would suggest to report age as median and interquartile range. 

We have reported age as median and interquartile range as suggested. Please see table 1 of the revised manuscript.

-> How do the participants know how many vaccination they have administered?

As suggested, we have provided more information about this issue. Please see p. 3 lines 114-116 of the revised manuscript.

-> Table 1: Suggest to include all abbreviations used in the table as a footnote

We have reported the extended name in table 1. Please see table 1 of the revised manuscript.

-> Are there any information on demographics of the participants?

We thank the reviewer for this comment. Participants included in the study were healthcare workers involved in vaccination services or general practitioners. Except for age and sex, no other demographic information was collected from the participants. We have added sex data of the participants in table 1 and S1 of the revised manuscript.

-> What does normative adherent mean?

We thank the reviewer for this comment. Those who fall into this category are the users who accepted the recommendations provided by the institutions (government and ministry of health) and get vaccinated on time without any particular doubts or concerns. For better clarity, we decided to change the name of this category into “users compliant to the recommendations”. Please see p. 4 lines 170, 178 of the revised manuscript.

Discussion

It would be helpful to discuss any unique findings relative to existing studies done in the country vs neighouring countries and any novel implications if any. 

We thank the reviewer for this comment, we have compared our results to other finding from different countries.  Please see the discussion section of the revised manuscript p.8 lines 420-423, 443-447.

Minor comments

A brief word related to the general high rates of COVID-19 vaccine hesitancy and especially among healthcare workers should be highlighted in the introduction. 

We thank the reviewer for this comment, the suggested articles have been useful to add a comment in our discussion about vaccine hesitancy among healthcare workers in Italy and in foreign countries. Please see p.8 lines 443-447 of the revised manuscript.

Round 2

Reviewer 2 Report

nil comments